# Cross-sectional study of association between socioeconomic indicators and chronic kidney disease in rural–urban Ghana: the RODAM study

David N Adjei,[1,2] Karien Stronks,[1] Dwomoa Adu,[3] Erik Beune,[1] Karlijn Meeks,[1] Liam Smeeth,[4] Juliet Addo,[4] Ellis Owusu-Dabo,[5] Kerstin Klipstein-Grobusch,[6,7] Frank P Mockenhaupt,[8] Ina Danquah,[9,10] Joachim Spranger,[11,12,13] Silver Bahendeka,[14] Ama De-Graft Aikins,[15] Charles Agyemang[1]

For numbered affiliations see end of article.

**Correspondence to**
Dr David N Adjei;
dna@chs.edu.gh

## ABSTRACT

**Objectives** Studies from high-income countries suggest higher prevalence of chronic kidney disease (CKD) among individuals in low socioeconomic groups. However, some studies from low/middle-income countries show the reverse pattern among those in high socioeconomic groups. It is unknown which pattern applies to individuals living in rural and urban Ghana. We assessed the association between socioeconomic status (SES) indicators and CKD in rural and urban Ghana and to what extent the higher SES of people in urban areas of Ghana could account for differences in CKD between rural and urban populations.

**Setting** The study was conducted in Ghana (Ashanti region). We used baseline data from a multicentre Research on Obesity and Diabetes among African Migrants (RODAM) study.

**Participants** The sample consisted of 2492 adults (Rural Ghana, 1043, Urban Ghana, 1449) aged 25–70 years living in Ghana.

**Exposure** Educational level, occupational level and wealth index.

**Outcome** Three CKD outcomes were considered using the 2012 Kidney Disease: Improving Global Outcomes severity of CKD classification: albuminuria, reduced glomerular filtration rate and high to very high CKD risk based on the combination of these two.

**Results** All three SES indicators were not associated with CKD in both rural and urban Ghana after age and sex adjustment except for rural Ghana where high wealth index was significantly associated with higher odds of reduced estimated glomerular filtration rate (eGFR) (adjusted OR, 2.38; 95% CI 1.03 to 5.47). The higher rate of CKD observed in urban Ghana was not explained by the higher SES of that population.

**Conclusion** SES indicators were not associated with prevalence of CKD except for wealth index and reduced eGFR in rural Ghana. Consequently, the higher SES of urban Ghana did not account for the increased rate of CKD among urban dwellers suggesting the need to identify other factors that may be driving this.

## INTRODUCTION

In general, individuals in lower socioeconomic status (SES) groups have been shown

### Strengths and limitations of this study

► The use of well-standardised study protocols across rural and urban Ghana eliminated intra protocol variability.

► Our study is also the first in Africa to use all three categories of chronic kidney disease (CKD) definitions (albuminuria, reduced estimated glomerular filtration rate and CKD risk) by Kidney Disease: Improving Global Outcomes 2012 in assessing association of socioeconomic status (SES) with CKD in rural and urban settings. This provides more detailed information on CKD outcomes.

► The limitation of intralaboratory variability in earlier studies was eliminated using the same standard operating procedures in the same laboratory for running all samples for both rural and urban Ghana.

► The use of three constructs of SES (educational level, occupational level and wealth index) in this study also provides a much better holistic approach to assessing SES associations with CKD.

► Our study was limited because of the use of cross sectional design which prevented us from determining causality between predictors and CKD progression.

to suffer more frequently from chronic kidney disease (CKD), often progressing to end-stage renal disease (ESRD), and associated with inadequate dialysis treatment, reduced access to kidney transplantation and poor health outcomes.[1] Recent studies have consistently found low SES to be associated with higher risk of CKD among people of African origin.[2–5]

However, in some settings the well-known inverse association between SES and CKD seems to be absent, or even reversed. For example, Bryne *et al* did not find any association between SES and ESRD.[6] Other studies have found a positive association between

SES and CKD.[7 8] Specifically, as SES improved, unhealthy lifestyle (unhealthy diet, physical inactivity, smoking and alcohol consumption) increased in China while that of the USA decreased with improved SES.[9] People with higher incomes, in these contexts, can afford a western lifestyle which is more readily available in the urban areas than in the rural areas. There is therefore an interaction between individual SES and environmental factors, such as food, alcohol, smoking and sedentary life style in such populations.[10–12] Consequently, in those settings, people with a higher SES might have higher CKD risk.

In urban areas, the population in general has higher SES than in rural areas.[13] For example, individuals with higher educational level migrate from rural areas to find higher occupations matching their higher education to improve on their wealth. If indeed a positive association between SES and CKD is observed in low/middle-income countries (LMICs), this might underlie the well-known health differences between urban and rural areas, with urban areas having an increased risk of CKD.[14] So far, it is unknown whether the reversed SES gradient (higher risk in high SES group) might explain the higher burden of CKD in urban areas as compared with rural areas in Africa.

In view of this, we assessed the association of SES with CKD in rural and urban Ghana and studied the extent to which the higher SES of people in urban areas could account for differences in CKD between rural and urban populations.

## METHODS
### Study population and study design
In the present analyses, data from the Research on Obesity and Diabetes among African Migrants (RODAM) study, a multicentre cross-sectional study were used. The rationale, conceptual framework, design and methodology of the RODAM study have been described in detail elsewhere.[15 16] As the Healthy Life in an Urban Setting study conducted among Ghanaian migrants living in Amsterdam did not find any associations between SES and CKD[17] the current study focused on rural and urban Ghana (Ashanti region of Ghana). The RODAM study was conducted from 2012 to 2015 and it comprised individuals aged 25–70 years living in rural and urban Ghana and Ghanaian migrants in Europe. All participants below 25 and above 70 years were excluded in the present analyses. The present analysis was restricted to the rural and urban sites (n=2492) RODAM participants. Specifically, 1043 participants from rural Ghana and 1449 from urban Ghana were used in this study.

Data collection for the study was standardised across all sites.

The response rate was 76% in rural Ghana and 74% in urban Ghana. In Ghana, participants were randomly drawn from a list of 30 enumeration areas in the Ashanti region based on the 2010 population census using the multistage random sampling. These enumeration areas came from two purposively selected urban cities (Kumasi and Obuasi) and 15 randomly selected rural communities in the Ashanti region. Selected health and community authorities were first identified, notified of the study and letters were sent giving detailed explanation of the study. We sent team members to stay among the communities to familiarise with them and organise mini clinics in the field. This lasted between 1–2 weeks depending on the sampled population and responsiveness of respondents.

In Ghana, questionnaires administration and physical examination were done at the same day/time. The participants were instructed to fast from 22:00 hours the night before the physical examination. For the current study, 2566 participants with data available on both questionnaire data and physical measurements were used. We excluded (n=74) individuals outside the RODAM age range of 25–70 years resulting in a data set of 2492 for analysis. These comprised 1449 Urban Ghana and 1043 Rural Ghana. For the final analysis, individuals with no data on CKD status (n=42) were excluded.

## MEASUREMENTS
### Covariates
#### Demographic and lifestyle factors
Information on demographics, educational level, occupational level, wealth index and lifestyle factors (smoking and physical activity) were obtained by questionnaire. Physical examinations were performed with validated devices per standardised operational procedures across all study sites. Weight was measured in light clothing and without shoes with SECA 877 scales to the nearest 0.1 kg. Height was measured without shoes with a portable stadiometer (SECA 217) to the nearest 0.1 cm. Body mass index (BMI) was calculated as weight (kg) divided by height squared ($m^2$). Overweight was defined as BMI of $\geq 25$ to $<30 \, kg/m^2$ and obesity as BMI $\geq 30 \, kg/m^2$.[18] Per participant, all anthropometrics were measured twice by the same assessor and the average of the two measurements were used for analyses.

### Predictor: SES
Socioeconomic indicators used in this study were educational level, occupational status and level of wealth index. Educational level was determined based on self-reported highest educational qualification accomplished based on the Ghanaian educational system. Occupational level was determined based on self-reported current occupation if employed and/or last occupation before retirement or student. The reported occupations were further coded according to the International Standard Classification of Occupations scheme (ISCO-08). Where 'high' (professionals, managers, clerical support staff, higher grade routine non-manual employees service and sales-related occupations) and 'low' (craft and related trades workers, elementary occupations and farmers) and the rest were categorised into the 'middle'. Wealth index was determined using the WHO standard of wealth index

**Table 1** Baseline characteristics by location

| | Rural Ghana | Urban Ghana |
|---|---|---|
| Number of participants, N (%) | 1043 (41.9) | 1449 (58.1) |
| Mean age, years (SD) | 46.5 (12.6) | 45.2 (11.4) |
| Females, N (%) | 638 (61.2) | 1034 (71.4) |
| Educational level n (%) | | |
| Low | 555 (56.9) | 614 (43.9) |
| Middle | 311 (31.9) | 547 (39.1) |
| High | 108 (11.2) | 239 (17.0) |
| Occupational status, n (%) | | |
| Low | 250 (25.7) | 374 (26.7) |
| Middle | 628 (64.5) | 818 (58.4) |
| High | 96 (9.8) | 209 (14.9) |
| Wealth index, n (%) | | |
| Low | 449 (46.5) | 368 (26.6) |
| Middle | 276 (28.6) | 416 (30.0) |
| High | 241 (24.9) | 602 (43.4) |
| BMI (kg/m$^2$) | | |
| <25 | 794 (76.3) | 579 (39.9) |
| 25–29.9 | 189 (18.2) | 495 (34.2) |
| ≥30 | 58 (5.5) | 374 (25.9) |
| Low physical activity, n (%) | 663 (47.2) | 592 (60.7) |
| Smoking, n (%) | 22 (2.3) | 14 (1.0) |
| Hypercholesterolaemia, n (%) | 78 (7.6) | 270 (18.7) |
| Hypertension, n (%) | 306 (29.3) | 531 (36.7) |
| Diabetes, n (%) | 53 (5.1) | 153 (10.6) |
| Albuminuria, n (%) | | |
| A1, normal to mildly increased (ACR <3 mg/mmol) | 930 (91.6) | 1285 (89.1) |
| A2-A3, moderately to severely increased (ACR ≥3 mg/mmol) | 85 (8.4) | 158 (10.9) |
| eGFR, n (%) | | |
| G1-G2 (≥60 mL/min/1.73 m$^2$) | 989 (96.3) | 1388 (96.3) |
| G3-G5 (<60 mL/min/1.73 m$^2$) | 38 (3.7) | 54 (3.7) |
| CKD risk, n (%) | | |
| Low risk (green) | 916 (90.5) | 1281 (88.9) |
| Moderately increased to very high risk (yellow to red) | 96 (9.5) | 160 (11.1) |

ACR, albumin–creatinine ratio; BMI, body mass index; CKD, chronic kidney disease; eGFR, estimated glomerular filtration rate; N, number of respondents.

classification. Wealth index was based on data collected in the Household Questionnaire. The questionnaire comprised questions on household's ownership of several consumer items such as television, car, flooring material, toilet facilities and so on. Each household was assigned a standard score for each asset. Wealth index was then expressed in five categories. The five categories were further categorised into three categories by combining the second and third as well as fourth and fifth categories due to small numbers.[19] All three SES constructs were further classified as low, medium and high SES and their relationship to each other tested. A composite SES variable (SES) was generated based on the three SES constructs (education, occupation and wealth index) using the EGEN group command in STATA V. 14.0. The codes were combined into numerical variables and their averages computed. The resultant values were recoded into three categories (low, medium and high).

### Comorbidity factors

Blood pressure (BP) was measured three times using a validated semiautomated device (The Microlife WatchBP home) with appropriate cuffs in a sitting position after at least 5 min rest. The mean of the last two BP measurements was used in the analyses. Hypertension was defined as systolic BP ≥140 mm Hg and/or diastolic BP ≥90 mm Hg and/or being on antihypertensive medication treatment and/or self-reported hypertension. Trained research assistants in the two sites collected fasting venous blood samples. All the blood samples were processed and aliquoted immediately (within 1 hour to maximum 3 hours of the vena puncture) after collection per standard operation procedures, and then temporarily stored at the local research location at −20°C. The separated samples were then transported to the local research centres laboratories, where they were checked, registered and stored at −80°C. To avoid intralaboratory variability, the stored blood samples from the local research centres were transported to Berlin, Germany for biochemical analyses. Fasting plasma glucose concentration was measured using an enzymatic method (hexokinase). Type 2 diabetes was defined according to the WHO diagnostic criteria (fasting glucose ≥7.0 mmol/L and/or current use of medication prescribed to treat diabetes and/or self-reported diabetes).[20] Concentration of total cholesterol was assessed using colorimetric test kits. All biochemical analyses were performed using an ABX Pentra 400 chemistry analyser (ABX Pentra; Horiba ABX, Germany). Hypercholesterolaemia was defined as total cholesterol level ≥6.22 mmol/L. Serum creatinine concentration (in umol/L) was determined by a kinetic colorimetric spectrophotometric isotope dilution mass spectrometry–calibrated method (Roche Diagnostics). Biochemical analyses were subject to extensive quality checks including blinded serial measurements.

### Outcome: CKD prevalence

Participants were asked to bring an early morning urine sample for the analyses of albuminuria and creatinine levels. Urinary albumin concentration (in mg/L) was measured by an immunochemical turbidimetric method (Roche Diagnostics). Urinary creatinine concentration (in umol/L) was measured by a kinetic spectrophotometric method (Roche Diagnostics). Estimated glomerular filtration rate (eGFR) was calculated using the CKDEPI (CKD Epidemiology Collaboration) creatinine equation.[21] Urinary albumin–creatinine ratio (ACR; expressed in

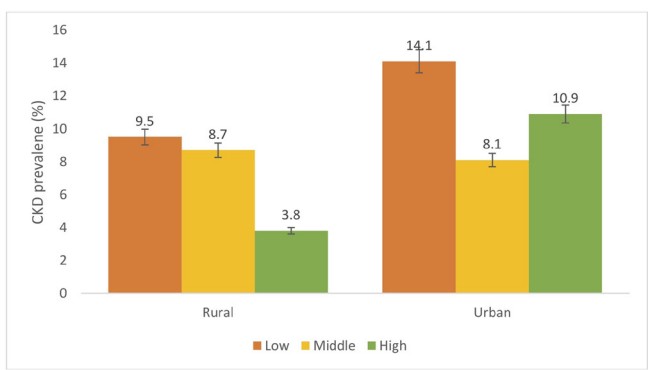

**Figure 1** Prevalence of chronic kidney disease (CKD) across level of education among urban and rural participants. Definitions according to 2012 Kidney Disease: Improving Global Outcomes guideline. CKD was defined as being in moderately increased risk, high-risk or very high-risk groups.

mg/g) was calculated by taking the ratio between urinary albumin and urinary creatinine. eGFR and albuminuria were categorised according to the 2012 Kidney Disease: Improving Global Outcomes (KDIGO) classification.[22] eGFR was categorised as follows: G1, $\geq$90 mL/min/1.73 m$^2$ (normal kidney function); G2, 60–89 mL/min/1.73 m$^2$ (mildly decreased); G3a, 45–59 mL/min/1.73 m$^2$ (mildly to moderately decreased); G3b, 30–44 mL/min/1.73 m$^2$ (moderately to severely decreased); G4, 15–29 mL/min/1.73 m$^2$ (severely decreased); and G5, <15 mL/min/1.73 m$^2$ (kidney failure). Albuminuria categories were derived from ACR and were as follows: A1, <3 mg/mmol (normal to mildly increased); A2, 3–30 mg/mmol (moderately increased); and A3, >30 mg/mmol (severely increased). CKD status was categorised according to severity of kidney disease (green, low risk; yellow, moderately increased risk; orange, high risk; and red, very high risk) using the combination of eGFR (G1-G5) and albuminuria (A1-A3) levels defined by the 2012 KDIGO guideline.[23] Due to the small number of participants in the very high risk category of CKD, high and very high risk groups were combined. Reduced eGFR was defined as eGFR <60 mL/min/1.73 m$^2$. Because of the small

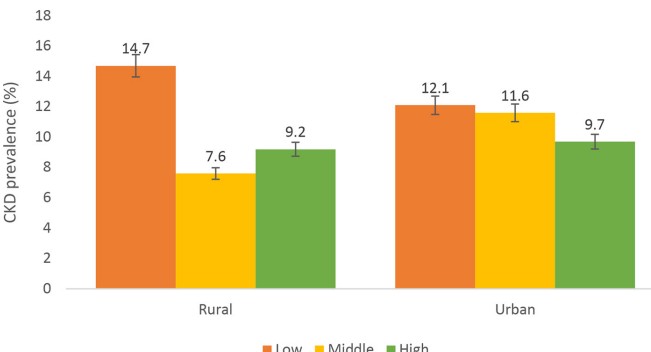

**Figure 2** Prevalence of chronic kidney disease (CKD) across occupational status among urban and rural participants. Definitions according to 2012 Kidney Disease: Improving Global Outcomes guideline. CKD was defined as being in moderately increased risk, high-risk or very high-risk groups.

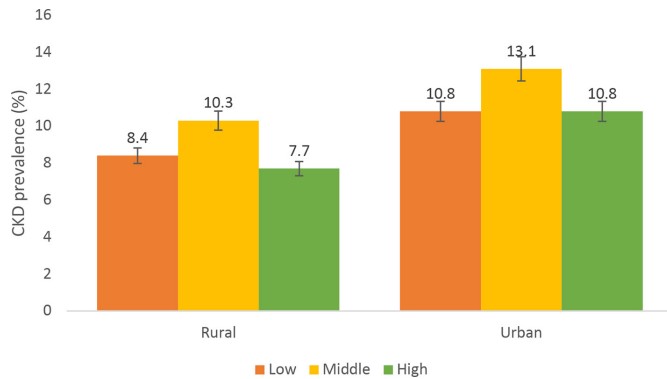

**Figure 3** Prevalence of chronic kidney disease (CKD) across wealth index categories among urban and rural participants. Definitions according to 2012 Kidney Disease: Improving Global Outcomes guideline. CKD was defined as being in moderately increased risk, high-risk or very high-risk groups.

number of participants in the severely increased albuminuria category, we defined albuminuria as ACR $\geq$3 mg/mmol by combining the moderately increased (A2) and severely increased (A3) categories.

### Patient and public involvement

Community leaders were involved in the recruitment of patients. These comprised religious communities (churches and mosques), endorsement from local key leaders and establishing relationships with healthcare organisations. We also provided information on the study by involving the local media (radio and television stations). We sent letters to all selected health and community authorities to notify participants of the study. Team members were sent to the various communities to stay among the community and organise mini clinics for a period of 1–2 weeks. Results of the study were disseminated through seminars, durbars and via radio and television stations.

### Statistical methods

Participants' characteristics were expressed as absolute numbers and percentages for categorical variables and as means and SD for continuous variables. CKD prevalence with 5% error bars were presented as bar graphs for each SES construct across rural and urban Ghana. Spearman's rank correlation was used to determine correlations between the three SES constructs. ORs and their corresponding 95% CIs were estimated by means of logistic regression analyses to study the odds of albuminuria (ACR >3 mg/mmol, A2-A3, moderately to severely increased albuminuria), reduced kidney function (eGFR <60 mL/min/1.73 m$^2$, G3-G5 moderately to severely decreased kidney function) and increased CKD risk (high and very high CKD risk) by SES, with adjustments for potential confounders (age and sex).[24] In addition, the analyses were performed for the total population (using low educational level, low occupational status and low level of wealth index as reference categories). Further analysis was conducted to assess the contribution of SES indicators to rural–urban differences in albuminuria,

**Table 2** Relationship between SES constructs (educational, occupational level and wealth index) by urban rural Ghana

| Correlation matrix | Educational level | Occupational level | Wealth index | SES |
|---|---|---|---|---|
| *Whole group* | | | | |
| Educational level | 1.000 | | | |
| Occupational status | **–0.060** | 1.000 | | |
| | 0.004 | | | |
| Wealth index | **0.282** | **–0.121** | 1.000 | |
| | 0.001 | 0.001 | | |
| SES | **1.000** | **–0.059** | **0.282** | 1.000 |
| | 0.003 | 0.006 | 0.001 | |
| *Urban Ghana* | | | | |
| Educational level | 1.000 | | | |
| Occupational status | **–0.115** | 1.000 | | |
| | 0.001 | | | |
| Wealth index | **0.294** | **–0.126** | 1.000 | |
| | 0.001 | 0.001 | | |
| SES | **1.000** | **–0.024** | **0.937** | 1.000 |
| | 0.002 | 0.001 | 0.001 | |
| *Rural Ghana* | | | | |
| Educational level | 1.000 | | | |
| Occupational status | 0.017 | 1.000 | | |
| | 0.589 | | | |
| Wealth index | **0.219** | **–0.135** | 1.000 | |
| | 0.001 | 0.001 | | |
| SES | **0.504** | 0.017 | **0.934** | 1.000 |
| | 0.001 | 0.587 | 0.001 | |

SES, socioeconomic status. Bold values are significant at 1%

reduced eGFR and CKD risk using rural Ghana as reference. Tolerance test and variance inflation factor showed very small degree of collinearity among SES predictors thus we therefore adjusted for each of SES variables separately. Complete case analysis approach was used. All data available were included in the age-adjusted models. All analyses were performed using STATA V.14.0.

## RESULTS

Table 1 shows characteristics of study participants. Participants in rural Ghana were slightly older than those in urban Ghana. Female preponderance was observed in both rural (61.2%) and urban (71.4%) Ghana, though higher proportions were observed in urban Ghana. Individuals living in rural Ghana were generally less educated (56.9%) compared with those living in urban (43.9%) Ghana. There were slightly more individuals with low occupational status in urban Ghana compared

with their peers in rural Ghana. People in urban Ghana (43.4%) were wealthier than their rural (24.9%) counterparts. Rural Ghanaians (47.2%) were more physically active compared with their urban peers. Smoking was low among Ghanaians though rural Ghanaians were more likely to smoke compared with their urban peers. Hypercholesterolaemia was more prevalent in urban Ghana than in rural Ghana. Hypertension (36.7%) and type 2 diabetes (10.6%) were more prevalent in urban Ghanaians compared with those living in rural Ghana. Urban Ghanaians were markedly more obese compared with their rural peers. Except for eGFR, albuminuria and CKD risk prevalence rates were higher in urban Ghana compared with rural Ghana.

Figure 1 shows prevalence of CKD by level of education in urban and rural Ghana. Prevalence of CKD decreased with increasing levels of education in rural Ghana. Higher prevalence of CKD was observed among individuals with low educational level compared with those with middle and high educational level. However, those with high educational level in urban Ghana had higher prevalence of CKD compared with those with middle level education. For occupational status, prevalence of CKD was higher among individuals with low occupational status in urban Ghana. Similar patterns were observed in rural Ghana; however, those with higher occupational status had higher prevalence of CKD compared with those with middle occupational status (figure 2). Figure 3 shows prevalence of CKD by level of wealth index. CKD prevalence among the levels of wealth index varied between urban and rural Ghana. Those with middle level wealth index had higher prevalence of CKD compared with those with low or high CKD prevalence in both rural and urban Ghana. CKD prevalence rate for low and high level wealth index in urban Ghana was the same while that of rural Ghana was slightly different.

Among the whole group, educational level was positively associated with wealth index (p<0.01) and composite SES (p<0.01). Occupational level was also inversely associated with educational level (p<0.01) and wealth index (p<0.01). In urban Ghana, high educational level was positively associated with high wealth index but inversely associated with occupation (p<0.01). In rural Ghana, high education was positively associated with high wealth index (p<0.01), but there was no significant association between education and occupation. High wealth index was inversely associated with high occupational status in both rural and urban Ghana (p<0.01) (table 2).

Table 3 shows association between level of education, occupational status, level of wealth index and prevalence of CKD. After adjusting for age and sex for the whole group, albuminuria was associated with middle level education (AOR=0.66, 0.48–0.91, p<0.01). After adjusting for age and sex, we observed no significant association between SES indicators (educational level, occupational status and wealth index) and CKD in urban Ghana. However, middle (AOR=0.51, 0.34–0.76,<0.01) and higher (AOR=0.53, 0.31–0.91, p<0.01) level education

**Table 3** Association of SES indicators (educational level, occupational status and wealth index level) with albuminuria, reduced eGFR and CKD risk

| | Albuminuria (ACR ≥3mg/mmol) | | eGFR <60mL/min/1.73 m2 | | High to very high CKD risk (KDIGO, 2012) | |
| | n (%) | OR (95% CI) Model 1 | n (%) | OR (95% CI) Model 1 | n (%) | OR (95% CI) Model 1 |
|---|---|---|---|---|---|---|
| **Education** | | | | | | |
| *Whole group* | | | | | | |
| Low | 1152 (11.89) | 1.00 (Reference) | 1160 (3.97) | 1.00 (Reference) | 1.150 (11.91) | 1.00 (Reference) |
| Middle | 847 (7.32) | **0.66 (0.48 to 0.91)** | 849 (3.77) | 1.36 (0.83 to 2.22) | 845 (8.28) | 0.82 (0.59 to 1.12) |
| High | 343 (7.00) | 0.67 (0.42 to 1.07) | 345 (3.19) | 1.11 (0.55 to 2.29) | 343 (8.75) | 0.96 (0.62 to 1.49) |
| *Urban Ghana* | | | | | | |
| Low | 612 (14.7) | 1.00 (Reference) | 612 (4.1) | 1.00 (Reference) | 612 (14.1) | 1.00 (Reference) |
| Middle | 546 (7.8) | **0.51 (0.34 to 0.76)** | 546 (3.7) | 1.12 (0.59 to 2.12) | 545 (8.1) | **0.59 (0.39 to 0.89)** |
| High | 238 (8.4) | **0.53 (0.31 to 0.91)** | 238 (3.4) | 0.91 (0.37 to 2.19) | 238 (10.9) | 0.83 (0.51 to 1.38) |
| *Rural Ghana* | | | | | | |
| Low | 540 (8.7) | 1.00 (Reference) | 548 (3.8) | 1.00 (Reference) | 538 (9.5) | 1.00 (Reference) |
| Middle | 301 (6.3) | 0.89 (0.51 to 1.59) | 303 (3.9) | 1.69 (0.77 to 3.66) | 300 (8.7) | 1.33 (0.79 to 2.25) |
| High | 105 (3.8) | 0.66 (0.23 to 1.95) | 107 (2.8) | 1.28 (0.35 to 4.71) | 105 (3.8) | 0.69 (0.23 to 2.02) |
| **Occupational status** | | | | | | |
| *Whole group* | | | | | | |
| Low | 614 (9.93) | 1.00 (Reference) | 616 (2.76) | 1.00 (Reference) | 613 (9.46) | 1.00 (Reference) |
| Middle | 1427 (9.25) | 0.82 (0.59 to 1.14) | 1436 (3.34) | 0.93 (0.52 to 1.66) | 1424 (9.90) | 0.89 (0.65 to 1.24) |
| High | 302 (10.26) | 0.76 (0.47 to 1.22) | 303 (7.92) | 1.33 (0.67 to 2.62) | 302 (12.91) | 0.90 (0.57 to 1.42) |
| *Urban Ghana* | | | | | | |
| Low | 207 (10.1) | 1.00 (Reference) | 207 (6.8) | 1.00 (Reference) | 207 (12.1) | 1.00 (Reference) |
| Middle | 817 (11.1) | 1.50 (0.88 to 2.83) | 817 (3.6) | 1.15 (0.56 to 2.35) | 816 (11.6) | 1.37 (0.84 to 2.56) |
| High | 373 (11.0) | 1.57 (0.89 to 2.53) | 373 (2.7) | 1.02 (0.41 to 2.52) | 373 (9.7) | 1.21 (0.68 to 2.14) |
| *Rural Ghana* | | | | | | |
| Low | 95 (10.5) | 1.00 (Reference) | 96 (10.4) | 1.00 (Reference) | 95 (14.7) | 1.00 (Reference) |
| Middle | 610 (6.7) | 0.65 (0.31 to 1.37) | 619 (3.1) | **0.37 (0.16 to 0.85)** | 608 (7.6) | 0.55 (0.28 to 1.08) |
| High | 241 (8.3) | 0.99 (0.43 to 2.28) | 243 (2.9) | 0.51 (0.18 to 1.44) | 240 (9.2) | 0.94 (0.44 to 2.01) |
| **Wealth index** | | | | | | |
| *Whole group* | | | | | | |
| Low | 808 (9.65) | 1.00 (Reference) | 813 (3.32) | 1.00 (Reference) | 808 (9.16) | 1.00 (Reference) |
| Middle | 678 (10.91) | 1.18 (0.84 to 1.66) | 683 (3.81) | 1.30 (0.74 to 2.28) | 675 (12.0) | 1.43 (1.02 to 2.01) |

Continued

**Table 3** Continued

| | Albuminuria (ACR ≥3mg/mmol) | | eGFR <60mL/min/1.73m2 | | High to very high CKD risk (KDIGO, 2012) | |
|---|---|---|---|---|---|---|
| | n (%) | OR (95% CI) Model 1 | n (%) | OR (95% CI) Model 1 | n (%) | OR (95% CI) Model 1 |
| High | 835 (8.62) | 0.93 (0.66 to 1.31) | 835 (4.19) | 1.55 (0.91 to 2.64) | 833 (9.96) | 1.21 (0.86 to 1.69) |
| *Urban Ghana* | | | | | | |
| Low | 367 (11.2) | 1.00 (Reference) | 367 (3.5) | 1.00 (Reference) | 367 (10.1) | 1.00 (Reference) |
| Middle | 414 (12.3) | 1.12 (0.73 to 1.74) | 414 (3.9) | 1.30 (0.61 to 2.80) | 413 (13.1) | 1.45 (0.93 to 2.27) |
| High | 601 (9.8) | 0.82 (0.55 to 1.25) | 600 (3.8) | 1.13 (0.55 to 2.31) | 600 (10.8) | 1.11 (0.72 to 1.71) |
| *Rural Ghana* | | | | | | |
| Low | 441 (7.9) | 1.00 (Reference) | 446 (3.1) | 1.00 (Reference) | 441 (8.4) | 1.00 (Reference) |
| Middle | 264 (8.7) | 1.13 (0.65 to 1.98) | 269 (3.7) | 1.22 (0.52 to 2.84) | 262 (10.3) | 1.31 (0.77 to 2.25) |
| High | 234 (5.6) | 0.78 (0.40 to 1.53) | 235 (5.1) | **2.38 (1.03 to 5.47)** | 233 (7.7) | 1.16 (0.63 to 2.14) |

Model 1, adjusted for age and sex.

%, proportion of individuals with CKD among the various levels of SES constructs in rural and urban Ghana. Bold values are significant at 5%

ACR, albumin–creatinine ration; CKD, chronic kidney disease; eGFR, estimated glomerular filtration rate; KDIGO, Kidney Disease: Improving Global Outcomes; n, total number of individuals in the whole group, rural and urban Ghana among the various levels of SES constructs; SES, socioeconomic status.

was associated with reduced albuminuria in urban Ghana. Whereas educational level and occupational status were not associated with CKD prevalence, high wealth index was significantly associated with higher odds of reduced eGFR in rural Ghana (AOR=2.38, 1.03–5.47, p<0.01).

Table 4 shows the contribution of all three SES constructs to rural and urban CKD prevalence differences. The odds of albuminuria and CKD risk was significantly higher in urban Ghana compared with rural Ghana (p<0.01). The higher rate of CKD observed in urban Ghana was not explained by the higher SES of that population as compared with their rural counterparts.

## DISCUSSION
### Key findings
Our study findings show no association between all three SES constructs and the prevalence of CKD in both rural and urban Ghana except for wealth index in rural Ghana, with the risk of CKD being higher in the wealthier populations. The higher rate of CKD observed in urban Ghana could not be attributed to the higher SES of that population compared with their rural counterparts.

### Discussion of key findings
#### Association of SES with CKD in rural and urban Ghana
Our study did not find any significant associations between all three SES constructs and CKD among rural and urban Ghana except for wealth index in rural Ghana. The positive association observed between wealth index in rural Ghana may be due to several reasons. A comparison of the three SES constructs showed higher educational level to be associated with wealth index in both rural and urban Ghana but not occupational level. This seems to suggest that occupational level may not be adequately capturing the SES status of individuals living in these settings in relation to CKD. For example, Masthi *et al*, compared different SES scales in rural and urban India and concluded that Standard of Living Index (SLI) scale was more accurate for classification of SES in urban and rural settings.[25] Our finding is consistent with other studies,[6 26] which reported no association between SES and CKD in high-income countries and LMICs, but in contrast with other studies[2–4 27] that found positive associations between SES and CKD. The reasons for our current finding are unclear. However, it has been suggested that these inconsistent associations may be due to the varying pathways through which the effect of SES on health status is mediated. For example, at a given educational level marked ethnic differences have been reported. Additionally, similar differences were observed for wealth status at a given income level.[28–30]

#### Contribution of SES to observed CKD risk differences between rural and urban Ghana
We observed higher rates of CKD in urban Ghana compared with rural Ghana, as expected. The observed higher rates of CKD in our study were not explained by

**Table 4** Contribution of SES indicators to rural–urban differences in albuminuria, reduced eGFR and CKD risk

| | | OR (95% CI) Model 1 | OR (95% CI) Model 2 | OR (95% CI) Model 3 | OR (95% CI) Model 4 | OR (95% CI) Model 5 |
|---|---|---|---|---|---|---|
| **Albuminuria (ACR ≥3 mg/mmol** | | | | | | |
| Sites | n cases (%) | | | | | |
| Urban Ghana | 1443 (10.9) | **1.37 (1.03 to 1.81)** | **1.70 (1.25 to 2.31)** | **1.55 (1.15 to 2.10)** | **1.62 (1.18 to 2.19)** | **1.74 (1.27 to 2.38)** |
| Rural Ghana | 1015 (8.4) | 1.00 (Reference) | 1.00 (Reference) | 1.00 (Reference) | 1.00 (Reference) | 1.00 (Reference) |
| **eGFR <60 mL/min/1.73 m$^2$** | | | | | | |
| **Sites** | n cases (%) | | | | | |
| Urban Ghana | 1442 (3.7) | 1.27 (0.82 to 1.97) | 1.20 (0.76 to 1.89) | 1.18 (0.79 to 1.86) | 1.12 (0.70 to 1.78) | 1.07 (0.67 to 1.72) |
| Rural Ghana | 1027 (3.7) | 1.00 (Reference) | 1.00 (Reference) | 1.00 (Reference) | 1.00 (Reference) | 1.00 (Reference) |
| **High to very high CKD risk** | | | | | | |
| **Sites** | n cases (%) | | | | | |
| Urban Ghana | 1441 (11.1) | **1.23 (1.01 to 1.62)** | **1.44 (1.07 to 1.93)** | **1.38 (1.03 to 1.84)** | **1.36 (1.01 to 1.83)** | **1.40 (1.04 to 1.91)** |
| Rural Ghana | 1012 (9.46) | 1.00 (Reference) | 1.00 (Reference) | 1.00 (Reference) | 1.00 (Reference) | 1.00 (Reference) |

Model 1: adjusted for age and sex; model 2: adjusted for age, sex and education level; model 3: adjusted for age, sex and occupational status; model 4: adjusted for age, sex and wealth index; model 5: adjusted for age, sex, educational level, occupational status and wealth index.

%, proportion of individuals with CKD among urban and rural Ghana.; ACR, albumin–creatinine ratio; CKD, chronic kidney disease; eGFR, estimated glomerular filtration rate; KDIGO, Kidney Disease: Improving Global Outcomes; n, total number of individuals in rural and urban Ghana; SES, socioeconomic status.

the higher SES of that population as compared with their rural counterparts. Our results indicate that this is due to the lack of a clear difference in the SES distribution of rural and urban Ghana observed in this study, as well as to the lack of associations between SES and CKD. Consistent with our findings, in a study conducted in Northern Tanzania SES did not explain increased risk of CKD in urban Tanzania.[26] The lack of associations between SES and CKD could probably and partly be explained by the process of epidemiological transition in relation to the 'diffusion theory' of ischaemic heart disease mortality. This theory attributes the commencement of ischaemic heart disease to individuals in the high SES group due to their ability to afford behaviours (smoking, alcohol and sedentary lifestyles) which increased risk of ischaemic heart disease. The lower SES groups were later affected partially because of improved living standards, unhealthy life style imitation and urbanisation. The higher SES groups were the first to embrace behavioural changes required to decrease the risk of ischaemic heart disease and this resulted in reversing the gradient.[31] The rapid urbanisation of some rural communities in the Ashanti region of Ghana and the imitation of urban lifestyle could account for our finding. Also, it could be that whereas the

high SES group in urban Ghana has already embraced favourable behavioural changes, those in rural Ghana are yet to do so.[32] This explains the observed association of wealth index with CKD in rural Ghana but not in urban Ghana. Also, the interplay of other less understood or researched factors (eg, exposure to nephrotoxins, herbal medications, sepsis, psychosocial factors) may be influencing the pathway in which SES influences CKD prevalence and progression.

**Strengths and limitations**

Our study presents several strengths. First, we used well-standardised study protocols across rural and urban Ghana. Our study is also the first in Africa to use all three categories of CKD definition (albuminuria, reduced eGFR and CKD risk) by KDIGO 2012 in assessing association of SES with CKD in rural and urban setting, this provided more detailed information on CKD outcomes. The limitation of intra laboratory variability in earlier studies was eliminated using the same standard operating procedures in the same laboratory for running all samples for both rural and urban Ghana. The use of three constructs of SES in this study also provides a much better holistic approach to assessing SES. Also, the distribution

of SES in our study reflects on the national data allowing for generalisation of our findings. Our study was limited by the use of cross sectional design, which prevented us from determining causality between predictors and CKD progression. Furthermore, there were more women than men in our study due to the higher response rate in women compared with men. However, this applied to both rural and urban Ghana. We therefore do not expect this to influence our results in a significant way.

## CONCLUSION

All three SES constructs appear not to be associated with prevalence of CKD in urban and rural Ghana except for wealth index in rural Ghana. The observed higher prevalence of CKD in urban Ghana was not explained by the higher SES in urban Ghana. Our study seems to suggest that other non-traditional factors such as nephrotoxins, herbal medications, psychosocial stressors and misuse of over the counter drugs may play a role and underscores the need to further explore these factors.

**Author affiliations**
[1]Department of Public Health, Academic Medical Center, University of Amsterdam, Amsterdam Public Health Research Institute, Amsterdam, The Netherlands
[2]Department of Medical Laboratory Sciences, School of Biomedical and Allied Health Sciences, College of Health Sciences, University of Ghana, Accra, Ghana
[3]Department of Medicine, School of Medicine and Dentistry, University of Ghana and Korle-Bu Teaching Hospital, Accra, Ghana
[4]Department of Non-Communicable Disease Epidemiology, London School of Hygiene and Tropical Medicine, London, UK
[5]Department of Public Health, Kumasi Centre for Collaborative Research, KNUST, Kumasi, Ghana
[6]Julius Global Health, Julius Center for Health Sciences and Primary Care, University Medical Centre, Utrecht University, Utrecht, The Netherlands
[7]Division of Epidemiology & Biostatistics, School of Public Health, Faculty of Health Sciences, University of the Witwatersrand, Johannesburg, South Africa
[8]Institute of Tropical Medicine and International Health, Charité – University Medicine Berlin, Berlin, Germany
[9]Department of Molecular Epidemiology, German Institute of Human Nutrition Potsdam-Rehbrücke, Nuthetal, Germany
[10]Institute for Social Medicine, Epidemiology and Health Economics, Charité - Universitaetsmedizin Berlin, Berlin, Germany
[11]Department of Endocrinology and Metabolism, Charité-University Medicine Berlin, Berlin, Germany
[12]German Centre for Cardiovascular Research (DZHK), Berlin, Germany
[13]Center for Cardiovascular Research (CCR), Charité-University Medicine Berlin, Berlin, Germany
[14]MKPGMS - Uganda Martyrs University, Kampala, Uganda
[15]Department of Population studies, Regional Institute for Population Studies, University of Ghana, Legon, Ghana

**Acknowledgements** The authors are very grateful to the research assistants, interviewers and other staff of the five research locations who took part in gathering the data and the Ghanaian volunteers in all the participating RODAM sites. We gratefully acknowledge the advisory board members for their valuable support in shaping the RODAM study methods and the Academic Medical Centre Biobank for their support in biobank management and high-quality storage of collected samples.

**Contributors** My co-authors have all contributed substantially to this manuscript and approve of this submission. Research idea and study design: DNA, CA, KS, DA, EB, KM, JA; data acquisition and curation: DNA, CA, EB, KM, data analysis/ interpretation: DNA, CA, KS, DA, EB, KM, LS, JA, EOD, KKG, FPM, ID, JS, SB, ADA; statistical analysis: DNA, CA, KS. DNA, CA, KS, DA, EB, KM, LS, JA, EOD, KKG, FPM, ID, JS, SB, ADA contributed important intellectual content during manuscript

drafting or revision and accepts accountability for the overall work by ensuring that questions pertaining to the accuracy or integrity of any portion of the work are appropriately investigated and resolved. DNA and CA take responsibility that this study has been reported honestly, accurately and transparently; that no important aspects of the study have been omitted; and that any discrepancies from the study as planned have been explained.

**Funding** This work was supported by the European Commission under the Framework Programme (Grant Number: 278901). The Wellcome Trust supported Professor Smeeth's contribution, grant number WT082178. Professor Joachim Spranger was supported by the DZHK (German Center for cardiovascular research) and the Berlin Institute of Health (BIH).

**Disclaimer** The funders had no role in study design, data collection and analysis, decision to publish or preparation of the manuscript.

**Competing interests** None declared.

**Patient consent for publication** Not required.

**Ethics approval** The respective ethics committees in Ghana and the three European countries approved the study protocols before data collection began. Specifically, we obtained ethical clearance in Ghana from School of Medical Sciences/Komfo Anokye Teaching Hospital Committee on Human Research, Publication & Ethical Review Board. In the Netherlands, the Institutional Review Board of the AMC, University of Amsterdam gave approval for the study. In Germany, approval for the study was obtained from the Ethics Committee of Charite-Universitäts medizin Berlin. The London School of Hygiene and Tropical Medicine Research Ethics Committee gave approval for the study in the UK.

**Provenance and peer review** Not commissioned; externally peer reviewed.

**Data sharing statement** Data are available from the RODAM research cohort, a third party. EB affiliated with the RODAM research cohort and a co-author of this paper in accordance with the RODAM requirements for collaboration. EB is the Data Collection Coordinator of RODAM and may be contacted with further questions (e. j.beune@amc.uva.nl). Additionally, researchers interested in further collaboration with RODAM may see the following URL: http://www.rod-am.eu/

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
