## [Reviewer comments · BMJ Open]

ARTICLE DETAILS

TITLE (PROVISIONAL)	A CROSS-SECTIONAL STUDY OF ASSOCIATION BETWEEN SOCIOECONOMIC INDICATORS AND CHRONIC KIDNEY DISEASE IN RURAL-URBAN GHANA: THE RODAM STUDY
AUTHORS	Adjei Nana, David; Stronks, Karien; Adu, Dwomoa; Beune, Erik; Meeks, Karlijn; Smeeth, Liam; Addo, Juliet; Owusu-Dabo, Ellis; Klipstein-Grobusch, Kerstin; Mockenhaupt, Frank; Danquah, Ina; Spranger, Joachim; Bahendeka, Silver; De-Graft Aikins, Ama; Agyemang, Charles

VERSION 1 - REVIEW

REVIEWER	Fu ping West China Hospital of China, China
REVIEW RETURNED	20-Mar-2018

GENERAL COMMENTS	In order to find out which of high income country economic pattern and low income country economic pattern is applied to individuals living in rural and urban Ghana. The researchers used the baseline data from multi-centre Research on Obesity and Diabetes among African Migrants (RODAM) study to assess the association between Socio-Economic Status (SES) indicators and CKD in rural and urban Ghana, and found out that All three SES constructs appear not to be associated with prevalence of CKD in urban and rural Ghana except for wealth index in rural Ghana. PubMed, EMBASE, Cochrane Central Register of Controlled Trials was searched and several similar research was found: Socioeconomic disparities in chronic kidney disease. Nicholas SB1, Kalantar-Zadeh K2, Norris KC2. Adv Chronic Kidney Dis. 2015 Jan;22(1):6-15. doi: 10.1053/j.ackd.2014.07.002. CONTENTS : 1. (Abstract) all setting/ participants/outcome and measurements should be put together in to a subtitle of methods. 2.before the study conducted should there be a assessment of Ghana social economic level ?
---

REVIEWER	James Heaf Zealand University Hospital, Denmark
REVIEW RETURNED	14-May-2018

GENERAL COMMENTS

The authors have studied the relationship between socioeconomic status (SES) and CKD in rural and urban inhabitants of Ghana. SES indicators were not associated with prevalence of CKD except for wealth index and reduced eGFR in rural Ghana. Consequently, the higher SES did not account for the increased rate of CKD among urban dwellers.

Considering the general difficulties of performing studies in Africa, I believe that this high quality paper should be published after appropriate modifications. African studies on this topic are relatively rare, so this investigation is welcome.

The methodology is sound, and the paper well written.

1) The statistics are problematic. For each SES variable (and the composite SES), when comparing to the three CKD variables, a Spearman correlation analysis is to be performed, and R and p values presented. After this, individual comparisons between groups can be performed (three per SES variable), using e.g. Mann Whitney. If the Spearman is not significant, the latter are only hypothesis-generating. I doubt whether the data are strong enough to demonstrate U-shaped correlations.

2) For urban-rural comparisons, four analyses are possible: one the population as a whole, and one for each SES group (low, middle and high).

3) In addition to urban and rural analyses, an analysis of the two groups combined would also be useful.

4) There is a general lack of statistics in the paper. While detailed p values are not necessary, all statements should be accompanied by some p value (e.g. $p < 0.05$, $p < 0.01$, $p < 0.001$) either in the text, tables or figures. The statements in paragraph 10-19 to 10-37 in particular all need to be significant if they are to remain in the paper, with the caveat mentioned in above.

5) I do apologize, but I find Table 4 a bit difficult to comprehend (and it is central to the conclusion). "The higher rate of CKD observed in urban Ghana was not explained by the higher SES of that population as compared to their rural counterparts." Does this not require a Model 5, adjusted for age, sex, education, occupation and wealth (or composite SES index)? Or have I misunderstood something?

Minor Comments

1) Page 5, margin number 20 (5-20). The "Invariably" and "consistently" terms are difficult to comprehend. Consider deleting or clarifying.

2) Page 6-14. While an already published detailed population description does not need to be repeated, some more information is required. Recruitment was voluntary and not randomly selected. The populations are clearly not representative of the background population (too many women). Is there any data comparing the population to the background population?

3) 7-42. Quintiles are not used in this study. Please delete.

4) 7-3. Africans have a higher eGFR than Caucasians. Were there any Caucasians in the population? How many? Was race included in calculation of the CKDEPI eGFR?

5) Table 2. * means $p < 0.01$? Please specify.

6) Table 3. The correlation between Wealth and eGFR in rural Ghana is highlighted. Please highlight other significant correlations, e.g. Education and albuminuria in urban Ghana. If highlighting of significant values is to be used, this should be done in all tables.

7) A number of variables, e.g. BMI, waist circumference, are mentioned in the Methods, but not the Results. Either include them

	in the Results (Table 1) and/or the analysis, or delete any mention of them.
--	--

REVIEWER	Muchiri Wandai Health Systems Trust. South Africa
REVIEW RETURNED	25-May-2018

GENERAL COMMENTS	 [ ] On page 5 of 26, lines 10&11, there seem to be kind of repetition (...‘used were’ and again ...‘were used’). Consider ‘In the present analyses, data from the multi-centre cross-sectional RODAM (Research ...) study were used’. [ ] On page 6 of 26, line 8, the SES variables (education, occupation and wealth index) should be removed since their rightful place is in following section sub-headed ‘Covariates’. The subheading though can be something like ‘SES covariates’ since it only describes the SES variables. [ ] ‘Other variables’ look more like explanatory/independent variables in this study, and so could probably come before the ‘Outcome:...’ to follow the other explanatory variables (lifestyle and SES factors). Would change of subheading make it more meaningful/descriptive? E.g. ‘Other NCD biomarkers’ or ‘co-morbidity factors’. [ ] In Table 1, descriptive statistics for BMI and waist circumference not included. [ ] Table 2 (page 11 of 26), there is mention of a composite SES, which is not described anywhere how it was constructed. If constructed from the three SES variables, then it is obvious that it will have a strong/weak correlation with its constituents (e.g. 0.937 with wealth index). [ ] It is not clear since it is not mentioned anywhere, how the occupations were regrouped into 3 categories from a possible 10 groups according to the ISCO. I have feeling that there could be wrong labelling/coding (3=High, 2= Medium, 1=Low), while it should be vice versa. This could be cause of negative correlation between it and the other two SES variables. [ ] My biggest concern is with the 4 models, which I feel should be combined into one logistic model where the association of SES factors is assessed after adjusting for all other variables (urban/rural; lifestyle factors (smoking, physical activity and BMI); and co-morbidities (hypertension, diabetes and hypercholesterolemia) in addition to gender and age. This should have been done because most likely the distribution of all these factors differs by the SES variables. The effect of SES factors should be done in the same model NOT in different models.
---

VERSION 1 – AUTHOR RESPONSE

Reviewer: 1

CONTENTS :

Comment 1: (Abstract) all setting/ participants/outcome and measurements should be put together in to a subtitle of methods.

Response 1: In the abstract, all the sub-headings have been put together as methods. Pg. 1 Lines 49-54

Comment 2: Before the study conducted should there be an assessment of Ghana social economic level?

Response 2: Thank you for this question. The socio-economic level of Ghana is a well-known public knowledge. Its impact on health outcomes varies from one condition to the other. Whereas a few are known, most are unknown or poorly understood.

Reviewer: 2

The methodology is sound, and the paper well written.

Comment 1: The statistics are problematic. For each SES variable (and the composite SES), when comparing the three CKD variables, a Spearman correlation analysis is to be performed, and R and p values presented. After this, individual comparisons between groups can be performed (three per SES variable), using e.g. Mann Whitney. If the Spearman is not significant, the latter are only hypothesis-generating. I doubt whether the data are strong enough to demonstrate U-shaped correlations.

Response 1: Thank you for this comment. We have used spearman correlation analysis as recommended to show the direction and strength of correlation between all SES constructs as well as the composite SES as suggested. However, the use Mann-Whitney test to determine differences between SES was not conducted because it does not relate to the focus of this manuscript. Pg. 12 & 13.

Comment 2: For urban-rural comparisons, four analyses are possible: one the population as a whole, and one for each SES group (low, middle and high).

Response 2: We have conducted all four analyses as suggested by reviewer. Pg. 14 & 15

Comment 3: In addition to urban and rural analyses, an analysis of the two groups combined would also be useful.

Response 3: We have provided analysis of urban and rural combined as suggested. Pg. 14 & 15

Comment 4: There is a general lack of statistics in the paper. While detailed p values are not necessary, all statements should be accompanied by some p value (e.g. $p < 0.05$, $p < 0.01$, $p < 0.001$) either in the text, tables or figures. The statements in paragraph 10-19 to 10-37 in particular all need to be significant if they are to remain in the paper, with the caveat mentioned in above.

Response 4: Thank you for the suggestion. We have provided confidence intervals for all the tables which are more robust compared to p-values. In addition, we have indicated where p-values are significant by bolding the CIs included in the results section as well. Pg. 12-16.

Comment 5: I do apologize, but I find Table 4 a bit difficult to comprehend (and it is central to the conclusion). "The higher rate of CKD observed in urban Ghana was not explained by the higher SES of that population as compared to their rural counterparts." Does this not require a Model 5, adjusted for age, sex, education, occupation and wealth (or composite SES index)? Or have I misunderstood something?

Response: Thank you for pointing out this important omission. Tolerance test and variance inflation factor (VIF) showed very small degree of collinearity among SES predictors that was why we adjusted for each of the SES variables separately. This was indicated in the methods section. Nevertheless, the results remain same even after adjusting for all the SES indicators in one model. We have included a fifth model to address this. Pg. 16

Minor Comments

Comment 6: Page 5, margin number 20 (5-20). The “Invariably” and “consistently” terms are difficult to comprehend. Consider deleting or clarifying.

Response: The words “Invariably” and “consistently” have been deleted from the paragraph.

Pg. 4 Lines 138-139.

Comment 7: Page 6-14. While an already published detailed population description does not need to be repeated, some more information is required. Recruitment was voluntary and not randomly selected. The populations are clearly not representative of the background population (too many women). Is there any data comparing the population to the background population?

Response: We thank the reviewer for this point. The recruitment was indeed random. The reason why the number of women is more than men is because the response rate is higher in women than in men. This is also already discussed in previous RODAM study. We have now emphasized this in the limitation. Pg. 5 Line 186 and Pg. 18 Lines 442-444.

Comment 8: 7-42. Quintiles are not used in this study. Please delete.

Response: The word “quintiles have been deleted from the paragraph. Pg. 6 Lines 220-225

Comment 9: 7-3. Africans have a higher eGFR than Caucasians. Were there any Caucasians in the population? How many? Was race included in calculation of the CKDEPI eGFR?

Response: There were no Caucasians among the study participants from Ghana. We used CKDEPI definition which takes into account race.

Comment 10: Table 2. * means $p < 0.01$? Please specify.

Response: Table 2 has been modified to bring clarity to the use of *. Pg. 12-13.

Comment 11: Table 3. The correlation between Wealth and eGFR in rural Ghana is highlighted. Please highlight other significant correlations, e.g. Education and albuminuria in urban Ghana. If highlighting of significant values is to be used, this should be done in all tables.

Response: Table 3 has been modified as suggested. We have highlighted all p-values, ORs and CIs in Table 2, 3 and 4 to show significance. Pg. 12-16.

Comment 12: A number of variables, e.g. BMI, waist circumference, are mentioned in the Methods, but not the Results. Either include them in the Results (Table 1) and/or the analysis, or delete any mention of them.

Response: Table 1 has been modified to include BMI while waist circumference has been removed in the methods section. Pg. 9-10; Pg. 6 Lines 206-208.

Reviewer : 3

Comment 1: On page 5 of 26, lines 10&11, there seem to be kind of repetition (...‘used were’ and again ...‘were used’). Consider ‘In the present analyses, data from the multi-centre cross-sectional RODAM (Research ...) study were used’.

Response: We have modified the sentence on page 5 as suggested. Pg. 5 Lines 165-167.

Comment 2: On page 6 of 26, line 8, the SES variables (education, occupation and wealth index) should be removed since their rightful place is in following section sub-headed 'Covariates'. The subheading though can be something like 'SES covariates' since it only describes the SES variables.

Response: Thank you for this comment. We have moved SES variables description to the following section as suggested. Pg. 6 Lines 209-226.

Comment 3: 'Other variables' look more like explanatory/independent variables in this study, and so could probably come before the 'Outcome:...' to follow the other explanatory variables (lifestyle and SES factors). Would change of subheading make it more meaningful/descriptive? E.g. 'Other NCD biomarkers' or 'co-morbidity factors'.

Response: We have provided a subheading to bring clarity to the other variables indicated in the methods section. Pg. 7 Lines 226-245.

Comment 4: In Table 1, descriptive statistics for BMI and waist circumference not included.

Response: We have modified table 1 to include BMI and waist circumference has been deleted from the methods section as indicated. Pg. 6 Lines 204-208.

Comment 5: Table 2 (page 11 of 26), there is mention of a composite SES, which is not described anywhere how it was constructed. If constructed from the three SES variables, then it is obvious that it will have a strong/weak correlation with its constituents (e.g. 0.937 with wealth index).

Response: We have described how the composite SES was generated in the methods section. Pg. 6 Lines 223-225.

Comment 6: It is not clear since it is not mentioned anywhere, how the occupations were regrouped into 3 categories from a possible 10 groups according to the ISCO. I have feeling that there could be wrong labelling/coding (3=High, 2= Medium, 1=Low), while it should be vice versa. This could be cause of negative correlation between it and the other two SES variables.

Response: Thank you for your observation. We have provided how the occupational levels were grouped into the three categories and cross checked for wrong labelling. No wrong labelling was identified. Pg. 6 Lines 214-216.

Comment 6: My biggest concern is with the 4 models, which I feel should be combined into one logistic model where the association of SES factors is assessed after adjusting for all other variables (urban/rural; lifestyle factors (smoking, physical activity and BMI); and co-morbidities (hypertension, diabetes and hypercholesterolemia) in addition to gender and age. This should have been done because most likely the distribution of all these factors differs by the SES variables. The effect of SES factors should be done in the same model NOT in different models.

Response: Thank you for this suggestion. Indeed, this was considered at the early stages of the manuscript and because the conventional risk factors are in the pathway of the relationship between SES and CKD adjusting for them could take away SES effect. However, we have added a 5th model adjusting for age, sex and all three SES constructs. Pg. 15.

VERSION 2 – REVIEW

REVIEWER	James Heaf Zealand University Hospital Denmark
REVIEW RETURNED	22-Jul-2018

GENERAL COMMENTS	The authors have responded satisfactorily to previous comments. I have nothing further to add.
--

REVIEWER	Muchiri Wandai Health Systems Research section of the Health Systems Trust, South Africa
REVIEW RETURNED	21-Aug-2018

GENERAL COMMENTS	1. Pp 7 (lines 226 and 227): the calculation of the composite SES states the tool used for calculation but not the HOW (Is it average of the value labels (3=high, 2=medium, 1=low?) from the 3 SES constructs (education, occupation and wealth index.) 2. Table 3, if I guess correctly reports results for logistic regression testing the effects of the SESs (education, occupation and wealth index) on the outcomes (ACR, eGFR and CKD) separately for the whole population and also for the sub-populations (Urban/rural divide). On the other hand Table 4 reports the urban/rural effect on the outcomes after progressively adjusting for age, gender and SES. It should therefore be clear that the Model 1 reported on Table 3 is not the same as that on Table 4. On Table 3, the 2nd last column heading (n cases (%)) is not consistent with other subheadings (n (%)) in columns 2&4. The 'cases' may need to be deleted. 3. Minor issue: Below Table 4 (small prints), n is given two different meanings; n= number of participants, and n=total number of individuals in rural and urban Ghana. In the same place, the R in ACR should be ratio NOT ration 4. The authors in the discussion section note the following between lines 419 and 422: 'The complexities of influence of SES on prevalence and progression of CKD and the differential prevalence of established risk factors (diabetes, obesity and hypertension) in rural and urban Ghana may also contribute to the different associations of SES with CKD prevalence observed in rural and urban Ghana'. The risk factors mentioned here should have been accounted for in the logistic models since the authors acknowledged these to be strong confounders. This I had suggested it be considered in the first round of review as it could change the reported results.
---

VERSION 2 – AUTHOR RESPONSE

Reviewer: 1

Comment 1: Pp 7 (lines 226 and 227): the calculation of the composite SES states the tool used for calculation but not the HOW (Is it average of the value labels (3=high, 2=medium, 1=low?) from the 3 SES constructs (education, occupation and wealth index.)

Response 1: We have included a section on how the composite variable (SES) was generated. Pg. 7, Lines 223-226.

Comment 2: Table 3, if I guess correctly reports results for logistic regression testing the effects of the SESs (education, occupation and wealth index) on the outcomes (ACR, eGFR and CKD) separately for the whole population and also for the sub-populations (Urban/rural divide). On the other hand, Table 4 reports the urban/rural effect on the outcomes after progressively adjusting for age, gender

and SES. It should therefore be clear that the Model 1 reported on Table 3 is not the same as that on Table 4.

Response 2: We have distinguished model 1 in Table 3 from that on Table 4 as suggested. Pgs. 8,9, Lines 285-290. Page 14, 15.

Comment 3: On Table 3, the 2nd last column heading (n cases (%)) is not consistent with other subheadings (n (%)) in columns 2&4. The 'cases' may need to be deleted.

Response: We have amended Table 3 heading. Pg. 13,14.

Comment 4: Minor issue: Below Table 4 (small prints), n is given two different meanings; n= number of participants, and n=total number of individuals in rural and urban Ghana. In the same place, the R in ACR should be ratio NOT ration.

Response: We thank the reviewer for this comment. Authors have modified the small prints and corrected the footnotes accordingly. Pg. 15.

Comment 5: The authors in the discussion section note the following between lines 419 and 422: 'The complexities of influence of SES on prevalence and progression of CKD and the differential prevalence of established risk factors (diabetes, obesity and hypertension) in rural and urban Ghana may also contribute to the different associations of SES with CKD prevalence observed in rural and urban Ghana'. The risk factors mentioned here should have been accounted for in the logistic models since the authors acknowledged these to be strong confounders. This I had suggested it be considered in the first round of review as it could change the reported results.

Response 5: We thank the reviewer for this comment. We did not account for the established risk factors of CKD (diabetes, obesity and hypertension) because they are not confounders, but rather in the causal pathway. In addition, the main focus of the paper is to assess the contribution of SES to rural-urban differences in albuminuria, reduced eGFR and CKD risk, rather than contribution of established risk factors of CKD. We have removed the sentence 'The complexities of influence of SES on prevalence and progression of CKD and the differential prevalence of established risk factors (diabetes, obesity and hypertension) in rural and urban Ghana may also contribute to the different associations of SES with CKD prevalence observed in rural and urban Ghana' to prevent confusion and bring clarity to the discussion. Pg.17 line 415

VERSION 3 - REVIEW

REVIEWER	Muchiri Wandai University of the Witwatersrand, South Africa
REVIEW RETURNED	20-Nov-2018

GENERAL COMMENTS	My issue again only on the statistics and around results of Table 2. If the authors have confirmed that the 3 SES variables are coded similarly (1=Low, 2=Middle, 3=High), are they concerned about the negative correlation between occupation and the other 2 variables (Education and Wealth index)? Simply taken, it implies for example that most participants with a high education level or a high wealth index have a lower occupation level, and this is probably unexpected. If there is a data coding problem whereby the occupation variable is coded as 1=High, 2=Middle, and 3=Low, then this could be a reason for the reported results on Table 2, which is erroneous. The results of Table 4 for model 5 would also be affected and
---

	could possibly change affecting the discussion and conclusions thereby
--	--

VERSION 3 – AUTHOR RESPONSE

Reviewer:3

Comment 1: My issue again only on the statistics and around results of Table 2. If the authors have confirmed that the 3 SES variables are coded similarly (1=Low, 2=Middle, 3=High), are they concerned about the negative correlation between occupation and the other 2 variables (Education and Wealth index)?

Simply taken, it implies for example that most participants with a high education level or a high wealth index have a lower occupation level, and this is probably unexpected. If there is a data coding problem whereby the occupation variable is coded as 1=High, 2=Middle, and 3=Low, then this could be a reason for the reported results on Table 2, which is erroneous. The results of Table 4 for model 5 would also be affected and could possibly change affecting the discussion and conclusions thereby.

Response 1: We thank the reviewer for additional comments. We have checked the data coding and there is no error. Indeed, the negative correlation between occupation and the other two variables (Education and Wealth) came up during the writing of this manuscript and was extensively discussed. Upon reflection, the negative correlation is surprising as the relationship between the three SES constructs in this setting does not usually follow the well-known positive relationship observed in other populations especially high income countries. In many parts of Ghana and other African countries especially rural and semi-urban areas this negative correlation is expected. For example, several graduates who are unemployed seek for other lower occupations (uber drivers, shop attendants, sales personnel) which are not commensurate with their educational levels creating a negative relationship between the two variables (occupation and educational level). In the case of wealth index, there are several people living in the rural and semi-urban areas who are farmers, street market men and women etc. who are within low occupations by definition and classification. However, these people have very high income and wealth (e.g. income from cash crops such as cocoa) compared with others with very high educational levels such as teachers, nurses and other professionals. This is common in many parts of Ghana and other African countries and explain the observed negative correlations.

VERSION 4 - REVIEW

REVIEWER	Muchiri Wandai University of the Witwatersrand South Africa
REVIEW RETURNED	03-Mar-2019

GENERAL COMMENTS	I had raised the issue below and I am comfortable with the authors' responses and clarification 'My issue again only on the statistics and around results of Table 2. If the authors have confirmed that the 3 SES variables are coded similarly (1=Low, 2=Middle, 3=High), are they concerned about the negative correlation between occupation and the other 2 variables (Education and Wealth index)? Simply taken, it implies for example that most participants with a high education level or a high wealth index have a lower occupation level, and this is probably unexpected.
--

	If there is a data coding problem whereby the occupation variable is coded as 1=High, 2=Middle, and 3=Low, then this could be a reason for the reported results on Table 2, which is erroneous. The results of Table 4 for model 5 would also be affected and could possibly change affecting the discussion and conclusions thereby'
--	--